# Urban Health: Assessment of Indoor Environment Spillovers on Health in a Distressed Urban Area of Rome

**Alessandra Battisti** [1],*[iD], **Livia Calcagni** [1][iD], **Alberto Calenzo** [1][iD], **Aurora Angelozzi** [2], **Miriam Errigo** [3], **Maurizio Marceca** [2] **and Silvia Iorio** [4]

1   Department of Planning, Design and Technology of Architecture, Sapienza University of Rome, 00196 Rome, Italy; liviacalcagni@gmail.com (L.C.); alberto.calenzo@gmail.com (A.C.)
2   Department of Public Health and Infectious Diseases, Sapienza University of Rome, 00185 Rome, Italy; aurora.angelozzi@uniroma1.it (A.A.); maurizio.marceca@uniroma1.it (M.M.)
3   Department of Social Sciences and Economics, Sapienza University of Rome, 00185 Rome, Italy; m.errigo93@gmail.com
4   Department of Medico-Surgical Sciences and Biotechnologies, Unit of History of Medicine and Bioethics, Sapienza University of Rome, 00185 Rome, Italy; silvia.iorio@uniroma1.it
*   Correspondence: alessandra.battisti@uniroma1.it; Tel.: +39-3397739471

**Abstract:** It is notable that indoor environment quality plays a crucial role in guaranteeing health, especially if we consider that people spend more than 90% of their time indoors, a percentage that increases for people on low income. This role assumes even further significance when dealing with distressed urban areas, vulnerable areas within cities that suffer from multiple deprivations. The community-based interdisciplinary research-action group of the University La Sapienza focused on a complex in the outskirts of Rome. The aim was to assess the correlations between architectural aspects of the indoor environment, socio-economic conditions, such as lifestyles and housing conditions, and eventually health outcomes. The intent of providing a comparative methodology in a context where official data is hard to find, led to the integration of social, health, and housing questionnaires with various environmental software simulations. What emerged is that underprivileged housing conditions, characterized by mold, humidity, unhealthiness, thermohygrometric discomfort, architectural barriers, and overcrowding, are often associated with recurrent pathologies linked to arthritis, respiratory diseases, and domestic accidents.

**Keywords:** indoor environmental quality (IEQ); indoor health; social determinants; multidisciplinary approach; regeneration strategies; spatial segregation



## 1. Introduction

Several studies on human exposure to indoor pollution, published over the last few years by the European Commission [1] and the World Health Organization [2], show that indoor environments can be a greater threat to health than outdoor environments. As stated in the Global Burden of Disease 2010 study, according to global estimates, indoor pollution represents the third leading cause of malaise and illness. This correlation also comes to light in more recent reports [3] that describe the quantity and quality of diseases related to the indoor environment and that show, for example, that 20% of European citizens suffer from asthma.

The salubrity of confined spaces becomes even more significant if we consider that people spend an average of 90% to 98% of their time indoors, depending on the location [4–8]. This average is bound to further increase during crisis events that force people to stay indoors, such as heatwaves, hurricanes, or pandemics. In order to bridge the technical and qualitative gap between pathologies and indoor space design, the WHO has drawn up a series of guidelines related to housing, aimed at guaranteeing the right to health, under the assumption that a healthy home strongly contributes to a general state of well-being: physical, mental and social [3].

The definition given by the WHO in 1942 [9] is difficult to achieve at the present time, marked as it is by an aging population, the consequent increase in chronic and infectious diseases, and by a new attitude towards illness, which is more and more perceived as a threat [10]. Over the years the limits of this concept have been widely criticized, but only recently has it been suggested to shift the emphasis to adaptability and self-management in coping with social, physical, and emotional challenges. This may be achieved through a bio-psycho-social approach to health that, for the first time, takes into consideration 'social determinants' that are systematically unequally distributed, notwithstanding their crucial influence on citizens' health [9]. Studies conducted by Antonovsky and Dilani as well as WHO reports follow this direction and prove the close correlation between the built environment and physical, mental, and social health. These concepts are reiterated in the Sustainable Development Goals set up by the UN 2030 Agenda in 2016, based on the integration of the three dimensions of sustainable development, environmental, social, and economic, as a prerequisite for eradicating poverty in all its forms.

The role of urban health has been emphasized by the awareness of the impact of the built environment on public health drawing international attention to the role that urban planning and architectural design should play in redefining urban fabric and indoor spaces. As early as 2012, the Lancet Commission Shaping Cities for Health underlined how urban health required a multisectoral approach. It drew attention to how much unhealthy lifestyles, inadequate diet, and reduced outdoor physical activity levels contribute to the increase of non-communicable diseases, especially obesity, diabetes, and cardio-respiratory diseases. With reference to individual lifestyles, the PASSI Program (Progress of Health Authorities for Health in Italy) is a national surveillance program in public health based on the Behavioral Risk Factor Surveillance model adopted in many different countries and also introduced in Italy in 2006. It is an internal tool of the National Health System, as it is conducted by the Prevention Departments of the Local Health Authority, coordinated by the regions that make use of the technical-scientific support of the Istituto Superiore di Sanità. The questionnaire continuously collects, through sample surveys, information on the lifestyles and behavioral risk factors of the adult Italian population (18–69 years old) registered in the health registers lists, connected to the onset of chronic non-communicable diseases and on the degree of knowledge and adherence to the intervention programs that the country is implementing to prevent them [11].

In the last two decades, several theoretical and practical studies have focused on the topic of "cities and health". The MIT–AIA multi-year Health and Urbanism Initiative shows the correlations between the built environment and health and aims at identifying and activating effective strategies that are locally relevant and globally scalable. The Harvard Graduate School of Design and Harvard T.H. Chan School of Public Health led the Health and Places Initiative to investigate alternative models of urban development and indoor environments that have a positive impact on health and aging. Despite the special focus on office buildings, Elzeyadi's studies at the University of Oregon [12] develop an integrative and adaptive model for the definition of indoor comfort. The study starts with the assumption that current bio-physical theories of indoor comfort, such as thermal and visual comfort, adopt a deterministic environmental perspective that highlights only the role of the physical environment in designing healthy and comfortable confined spaces. These theories define indoor comfort in terms of independent components that affect the occupant's overall perception of comfort. Elzeyadi shows that, on the contrary, the occupant reacts to the overall atmosphere that results not only from the direct effect of each comfort parameter but also from the effect of their interaction. The perception of indoor comfort must therefore be considered a multisensorial experience that includes five subsystems: thermal comfort, visual comfort, indoor air quality, acoustic comfort, and spatial comfort.

Studying the parameters that improve living conditions is crucial not only to ensure inhabitants' good health, but also to maximize the performance of their indoor activities: sleep, study, work, relaxation, and socialization. Therefore, improving the housing condi-

tions of the most vulnerable households or communities, which are more likely to reside in inadequate housing, also contributes to their socio-economic empowerment [13]. The link between social position and health, defined by Marmot as the social gradient in health, was the main theme of the Commission on the Social Determinants of Health, set up by the WHO General-Director in 2005. After 3 years, in August 2008, the Commission published a final report, entitled "Closing the gap in a generation: health equity through action on the social determinants of health" [14]. At the heart of the report was the imperative for all governments to act on the social determinants of health in order to eliminate health inequalities between countries and within countries. Marmot himself, who had been a member of the Commission, highlights how social exclusion and dis-empowerment contribute to creating a social gradient in the state of health. A high level of cohesion and social participation, responsibility, good mobility, walkability, access to healthy food, adequate service standards, and green spaces and a general sense of security, all contribute to improving health. Such a high-quality urban system can allow and promote social interaction [13].

If it is true that socio-economic status is one of the most important predictors of morbidity and premature mortality at a global level, it is also true that the literature has highlighted the need to consider a wide range of aspects in the genesis of health inequalities [15]. Thanks to the increasingly strong connection between urbanism, architecture, anthropology, sociology, and medicine, we are now reaching an "ecological" health perspective [16,17].

The case study under consideration concerns the regeneration of the Ex Bastogi complex, located in the North-West of Rome, in the 13th municipality (Figure 1). The complex, consisting of six buildings erected in the 1980s as a residence for Alitalia employees, was subsequently assigned by the municipality of Rome to deal with the housing emergency, with the aim of assigning homes to families in need. These housing assignments were offered on a temporary basis (a few months at most), pending final assignment in permanent housing, under the Italian acronym ERP (Public Residential Housing) [18]. Although designed to accommodate 1100 users for short stays, today it has a stable population of 1033 inhabitants, according to the 2011 Istat census, and about 2000 according to qualitative studies (Figure 2) [19]. Overcrowding of the buildings, inadequate accommodation, and the perception of a transience condition extended over the years without any hope of being transferred by the municipality to the ERP, are all factors that contribute to worsening the condition of social hardship of the inhabitants. Although a pronounced social vulnerability can be observed in several urban areas around the world, the Ex Bastogi complex stands out due to the fact that the buildings' legal definition and status—to this day, about 30 years after its foundation—is devoid of a definitive framework [18]. The singularity of Ex Bastogi lies in the impossibility of ascribing it to any legal status. Indeed, the complex does not fall within the classification of Temporary Housing Assistance Centres (CAAT) or within the occupied buildings or the permanent housing projects (ERPs). Furthermore, Ex Bastogi is an enclave within the urban fabric of Rome that hosts various types of vulnerable groups: children and elderly people in a state of neglect, illegal migrants, people under house arrest, drug addicts, alcoholics, and people with HIV [18].

The study is part of a broader research-action project characterized by a community-based approach and conducted by a multidisciplinary research group of the Sapienza University of Rome in collaboration with other social, health and local institutions, including the Local Health Unit (ASL Roma 1), the Department of Epidemiology of the Health Regional System of Lazio (DEP), the XIII Municipality of Rome, the National Institute for the Promotion of the Health of Migrant Populations and the Contrast of Diseases of Poverty (INMP) aimed at investigating the effect of the unequal distribution of social determinants on the health of a population characterized by poor housing and socio-economic vulnerability.

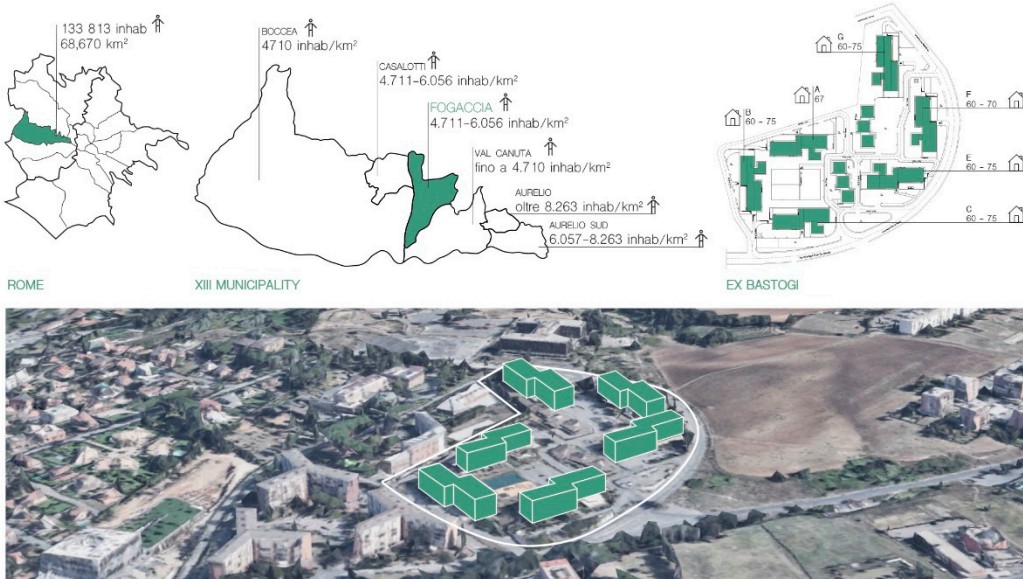

**Figure 1.** Territorial framework.

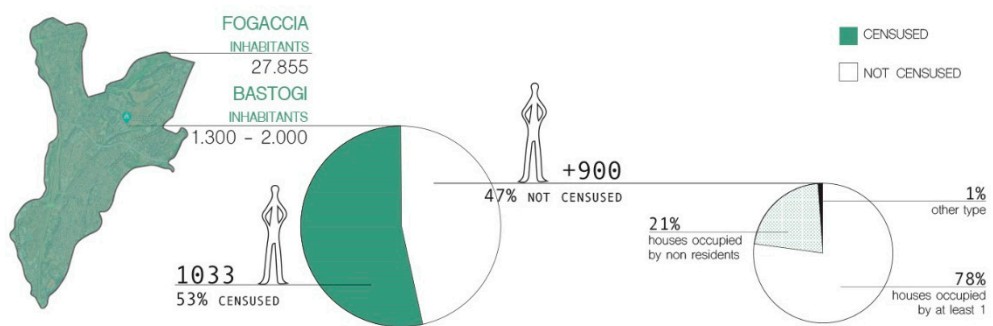

**Figure 2.** Estimated population, censused and not.

The general intention of the project is to put together an infrastructure of analysis and prototypical intervention that facilitates empirical studies aimed at adequately grasping the complexity of the relationships between health disparities, socio-environmental and economic discomfort, personal and collective health, involving the population and institutions in this process of analysis and implementation proposals [17]. The intent of the study is to assess the impact that social gradients can have on the lifestyle and well-being of a population, especially in such a marginal area, where housing conditions strongly contribute to an increase in mortality and morbidity amongst the most vulnerable and discriminated population groups.

The awareness that Ex Bastogi was a complex system led to a preliminary assessment of the social, environmental, and architectural conditions, in order to achieve a broad, intersectoral, interdisciplinary, and holistic understanding of the context [20]. Barton's Settlement Health Map [21] was used as an analytical tool and as a starting point for the research. In the map, health and well-being are placed at the center, while the social, economic and environmental aspects of a settlement are arranged around it in concentric circles [18].

The researchers carried out a participatory research-action plan, focusing on improving housing conditions. This was a significant and relevant issue for the inhabitants themselves, who were involved from the very stage in the entire process. In order to develop a sense of community and social cohesion, the regeneration process of the Ex Bastogi center began with the creation of outdoor common spaces for socialization, thus improving bio-psycho-social well-being and helping to fight against inequities. The research project then focused on the housing conditions with a view to regeneration. This phase was preceded by

community activities, such as Focus Groups, questionnaires, and community consultations with some members of the local population. This community-based approach allowed us to combine qualitative and quantitative analysis for the regeneration project.

## 2. Materials and Methods

The aim of this study is to combine the architectural analysis, concerning spatial comfort, indoor discomfort, accessibility, usability, and safety, with the results from the data collected through two questionnaires:

- PASSI questionnaire (Progressi delle Aziende Sanitarie per la Salute in Italia);
- Social, housing, and cognitive questionnaire on the residents' access to social and health services.

Given the case study area, a selection of questions of the PASSI questionnaire were taken into consideration: item 1.5 in the Health condition and perceived quality of life session, which investigated the diagnosis of one or more pathologies diagnosed by the interviewee; item 3.3 and 3.10 in the Smoking habits session, which investigated, respectively, if one is a current smoker and if one smokes at home, items 13.1 and 13.1b in the Home safety session, which investigated the possible admission to the emergency room for domestic accidents and the perception of the domestic accidents risk.

The second questionnaire focused on items 2.3 to 2.7, which investigate with whom one lives, house dimensions, problems detected in the dwelling, heating efficacy, and obstacles that can limit the access of individuals with movement problems.

### 2.1. Sampling and Data Collection

For the purposes of this study, a probabilistic, simple random sampling with correction for finite populations was used. As per the PASSI protocol, the sample, consisting of 250 subjects, was extracted from a population of 642 people (Ex Bastogi inhabitants, aged between 18 and 69 years and registered in the ASL of competence health register).

Because of the difficulties encountered in administering the questionnaires, due to distrust and lack of participation, 210 people were reached from a sample of 250 subjects (response rate: 84%). Of the 210 people interviewed, 134 (63.8%) were female and 76 (36.2%) were male; in particular, 65 people (of which 38 females and 27 males) were aged between 18 and 34 years, 57 people (of which 41 females and 16 males) were between 35 and 50 years old, 88 people (of which 55 females and 33 males) were between 51 and 69 years old.

The interviews were mainly led in person and, only in some cases, by telephone. Before administering the questionnaires, each researcher explained its content and purpose to the interviewee, informing them that anonymity would be ensured and that the data would be used exclusively for research purposes, as required by the national law on personal data processing [22].

The administration of the questionnaires began in the second half of March 2018 and ended in June 2019. When the respondent voluntarily interrupted the administration the questionnaire was excluded. All unanswered items were reported as empty responses. The data collected through the questionnaires were entered into a database, specifically created using Microsoft Access 2013 software. The data were then exported to Excel and cleaned, before carrying out the descriptive analysis.

### 2.2. Architectural, Energetic, and Environmental Analysis of Housing Conditions

In light of the methodology adopted in the scientific literature, the evaluation parameters chosen for the housing conditions analysis correspond with some of the priority areas addressed by the HHGL [3] to ensure health indoors: inadequate living space (crowding); low or high internal temperatures; risk of home injuries; housing accessibility for people with functional disabilities.

With regard to the analysis of housing conditions, the approach adopted was the one indicated and developed in the Harvard T.H. Healthy Buildings Program at the Chan School of Public Health, summarized in the "9 Foundations of a Healthy Building" [23]:

thermal comfort, humidity, ventilation, air quality, dust and parasites, safety, water quality, acoustic comfort, and, finally, visual comfort and lighting. These aspects are evaluated and compared through health performance indicators (HPIs) in order to provide guidelines for optimizing buildings in terms of salubrity. A further methodological contribution is provided by the Chair of Building Technology and Climate Responsive Design at the Technische Universität München [24], which explores the connections and interdependencies between indoor environment factors with spillovers into human health and human biological signals that activate a physiological reaction in response to environmental stressors. With regard to the first factors, the study highlights the variables that influence indoor air quality (IAQ), such as airborne contaminants, ventilation contribution, humidity level, and indoor environmental quality (IEQ), from a thermal, visual, and acoustic point of view, highlighting the direct correlations with health.

Human beings experience their habitat through their perceptual and metabolic system as a multisensory and multidimensional reality [25]. The sensations of physical well-being ensue from the overall response of the organism to the surrounding environment, combined with subjective cultural and emotional aspects [26]. Given this complex variety of interrelated factors, the present study has focused specifically on thermo-hygrometric and visual comfort (lighting).

Therefore, the parameters taken into consideration for the living conditions analysis are:

- spatial comfort;
- thermal comfort;
- visual comfort (natural and artificial lighting);
- safety, in the most inclusive meaning that embraces accessibility and living space [3].

### 2.3. Methodology and Tools for Data Collection

The architectural analyses were carried out based on the approved project, filed with the Municipality of Rome (Dipartimento XII, Lavori Pubblici e Manutenzione Urbana). All data was verified through several inspections in order to compare the filed project with the current situation (indoor common spaces, roofs, cellars, and number of apartments) as documented by metric and photographic surveys.

The thermal transmittance calculation was used to quantify heat losses through the existing building envelope. These results were then compared with those pertaining to a new more efficient building envelope. Reference was made to the UNI EN ISO 10077-1: 2018 (Thermal performance of windows, doors, and shutters—Calculation of thermal transmittance—Part 1: General information) and to the UNI EN ISO 10077-2: 2018 (Thermal performance of windows, blackout doors, and closures—Calculation of thermal transmittance—Part 2: Numerical method for frames). In the absence of complete information about the building envelope components, opaque horizontal closing elements in brick-cement and vertical closing elements in prefabricated concrete panels (with no type of thermal insulation)—as infill of the load-bearing structure in reinforced concrete—were hypothesized (Figure 3).

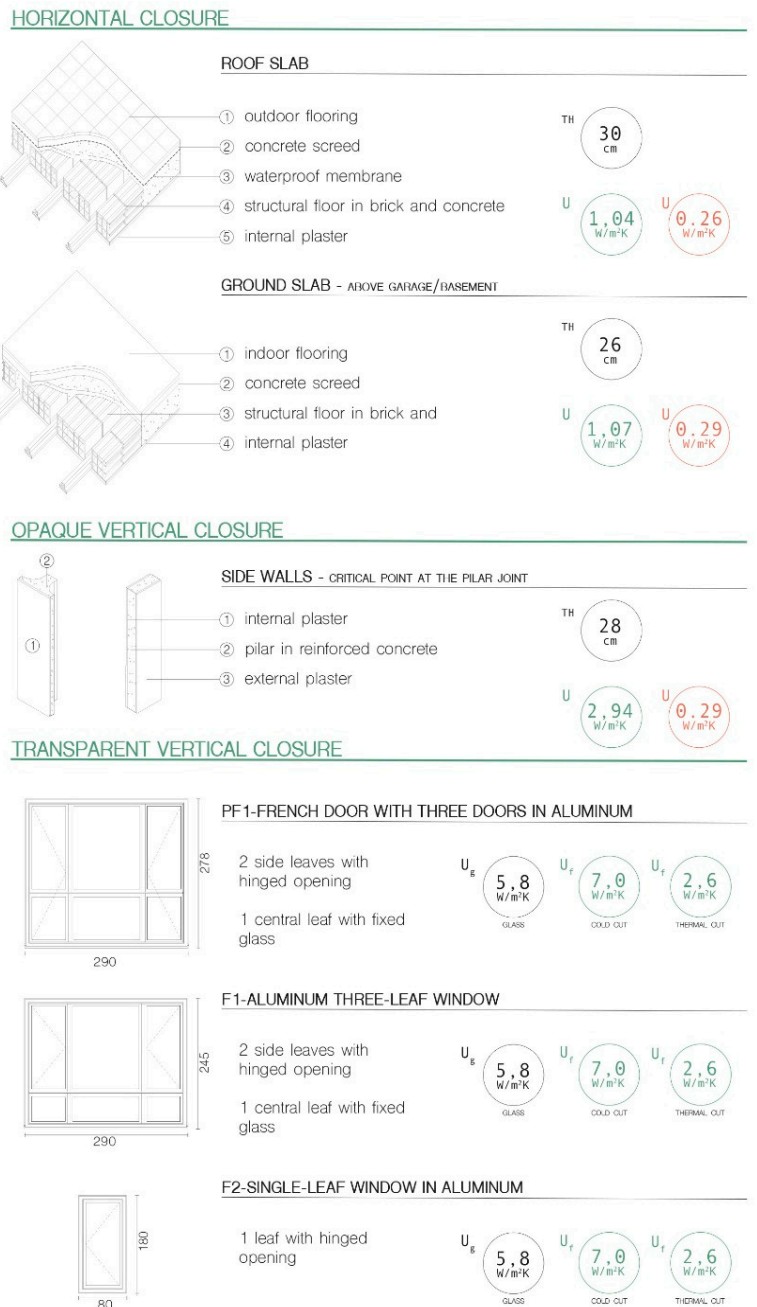

**Figure 3.** Hypothesized building envelope components.

With a view to elaborating a building regeneration project, some representative and recurring housing units were selected as the object for the simulations that concerned both the current situation and the situation following potential intervention strategies. The housing typologies chosen are those used on the standard floor of one of the six linear buildings of the complex, in particular, those with exposure to the NE and SW. The architectural study paved the way for environmental analysis. Software packages such as Ecotect, ENVI-met, Relux, and Flow Design were, respectively, used to evaluate solar exposure and solar gain, microclimatic conditions, natural and artificial lighting within some typological accommodations, and outdoor and indoor airflows. The identification of the context, according to the Koppen classification, is on the basis of the environmental and energetic analysis. The area is characterized by the Mediterranean climate Csa, which represents a particular sub-category of subtropical climate. The current condition and the

condition following potential intervention strategies were then assessed through a type of calculation software that could compare the existing scenario with an improved one.

- **ENVI-met**. The biometeorological and microclimatic conditions of the area, studied both during summer and winter seasons, were analyzed using the CFD software (Computational Fluid Dynamic) ENVI-met v.3.1, through the modeling of four systems: soil, buildings, vegetation, and atmosphere. The simulations were carried out on 21 June 2016 and on 21 December of the same year (from 08:00 to 20:00) as they were considered the most appropriate days to analyze the effects of materials and vegetation on the microclimate and thermal comfort conditions of the area under examination. The climate input data were set according to the default values provided by the software for the city of Rome following the geolocation of the model. The setting was approximated in terms of spatial configurations within the software-specific constraints according to the following settings:
  - Simulation box with a grid of $400 \times 400$ m and a height of 35 m corresponding to about twice the buildings height to ensure a correct simulation;
  - Sizing of the cells that make up the grid in terms of surface and height equal to $2 \times 2 \times 2$ m;
  - Definition of the materials that characterize the buildings envelope, the surfaces, and the vegetation through the Database Manager, precisely:
    1. Building envelope: "Default Wall—moderate insulation" for roofs and perimeter walls;
    2. Soil and surfaces: "Asphalt" for roads, "Default unsealed soil" for ground, "Grass 25 cm average density" for green areas, "Concrete pavement light" for pavements and cemented areas;
    3. Trees: "Tree 15 m/20 m very dense, distinct crown layer";
- **Ecotect**. The sunlight analysis, aimed at evaluating natural lighting/shading and the incident solar radiation on different surfaces, was carried out using Autodesk Ecotect Analysis software. The climatic input data are related to the latitude and longitude of the city of Rome (Lat. 41.8°, Long. 12.6°) and allowed to define the actual solar path during the year. A prerequisite for this type of analysis was the creation of a three-dimensional oriented model of the six existing buildings and of their respective surroundings that define, though in a discretized way, the volumes and position of buildings, trees, and, more generally, any influential element useful for the analysis.

The simulations were carried out during the summer and winter solstices since they represent the most unfavorable conditions, in terms of solar input, for the respective season. The settings were as follows:

- Summer: June 21st with a simulation time range from 08:00 to 18:00
- Winter: December 21st with a simulation time range from 09:00 to 17:00

In order to evaluate the lighting and air flows of the indoor spaces, the analysis examined the 25 square meter single-facing apartments that constitute the basic module and recurs in different orientations.

- **Relux**. The 3D graphics software Relux was used to analyze indoor natural lighting. The first step was the insertion of basic data regarding indoor spaces dimensions, location of the evaluation areas of each room to be calculated: 25 square meter room exposed to SW; 25 square meter room facing NE. Regarding the calculation data, reference was made to the DIN EN 12464-1 values. The simulations, carried out during the summer and winter season, in particular on 21 June and 21 December at 13:00, were set as follows:
  - Calculation algorithm: Average indirect fraction
  - Height of evaluation surface: 0.75 cm
  - Calculation mode: CIE overcast sky

The software was also used for artificial indoor lighting analysis, assuming the presence of 75 W incandescent bulbs. The positioning and power of the lighting devices were assumed according to what was detected during the inspections: three devices were placed according to the needs and main activities. The layout and power of the devices may undergo slight variations in some modules. The simulations were set up as follows:

- Calculation algorithm: Average indirect fraction
- Reference plan: 0.75 cm
- Lamp type: Incandescent Lamp
- **Flow Design**: Flow Design software, a wind tunnel testing software that returns fluid dynamics simulations, was used to analyze air flows inside the modules. A discretized three-dimensional model of two housing typologies was created: single-facing and double-facing. The simulation settings were as follows:
  - Status: Transient
  - Analysis: 3D
  - Wind speed: 3 m/s
  - Voxel size: 0.436 m
  - Orientation: south-west and north-east
  - Flow lines: plane
  - Visualization of the wind speed: lines

## 3. Results

The outcomes of the questionnaires and the architectural analysis have both highlighted the marginalization condition that characterizes the complex. In fact, the area perfectly falls into the category that the OECD defines as "Distressed Urban Areas", situations of underdevelopment in developed contexts, that is, areas in which there are serious conditions of underdevelopment compared to the city itself and to the national average. This marginalization must be contextualized within the XIII Municipality, which is marked by an evident center-periphery gradient in the distribution of social determinants.

*3.1. Questionnaires*

The PASSI questionnaire, combined with the social and housing one, returned an alarming picture of the living conditions, characterized by the presence of mold, humidity, unhealthiness, thermal and hygrometric dis-comfort, architectural barriers, and overcrowding.

Concerning the level of education, 2% of respondents have no qualifications, 9% have obtained an elementary degree, 53% have obtained an average degree, 29% have a diploma or a secondary school qualification, and 6% have a degree.

At the time of the interview, 40.2% of the respondents stated that they were continuously employed and 73.1% stated that they had significant or some difficulties in coping with their normal monthly expenses.

With regard to health conditions, out of the 210 respondents, 21 subjects reported having had a diagnosis of asthma (10%). Out of these 21 subjects, only 4 are non-smokers and all 4 have problems with humidity and/or drafts in their home (19%). Out of 210 subjects, 9 have been diagnosed with COPD (4.3%). Out of these 9 respondents, only 3 are non-smokers and of these 3 non-smokers only 2 have problems with humidity and/or drafts at home.

Furthermore, out of 210 respondents, 18 have been diagnosed with osteoarthritis (8.6%). Of these 18 subjects, 8 of them complain about humidity and/or draft problems in the house (44.4%).

The presence of poor housing conditions can also increase the risk of domestic accidents. Out of 210 respondents, 27% consider the possibility of getting injured at home as high or very high.

*3.2. Spatial Comfort*

According to the approved project, filed with the Municipality of Rome (Dipartimento XII, Lavori Pubblici e Manutenzione Urbana), each of the six buildings of the Ex Bastogi complex consists of five floors above ground and comprising about 67 lodgings for about 190 users. The structural grid defines the repetition of the same 25 square meter module that forms the smallest accommodation and, when combined, constitutes larger houses from 50 to 120 square meters (Figure 4). The prevailing typology is the 43 square meters one equipped with a loggia or the 52 square meters one without loggia.

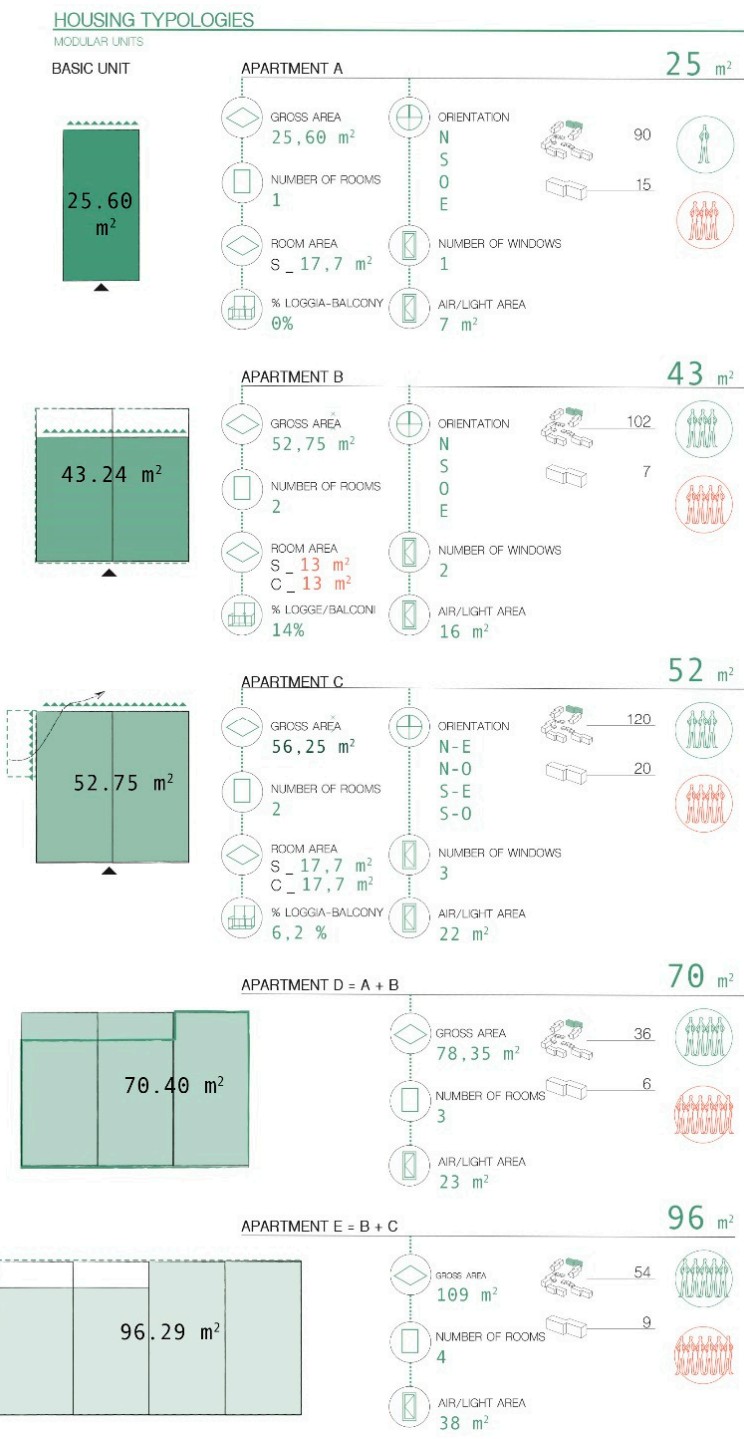

**Figure 4.** Housing typologies, modular units.

A comparison between the project data (number of dwellings and dimensional aspects) and the effectively settled inhabitants was made (Table 1).

**Table 1.** The number of assumed hosted inhabitants exceeds the number of censused ones by 213% and the number of inhabitants that can be hosted in compliance with the regulations by 190%.

| Hosted Inhabitants | Censused Inhabitants | Hostable Inhabitants |
| --- | --- | --- |
| 2200 | 1033 | 1158 |

### 3.3. Thermo-Hygrometric Comfort

The thermal loss calculation includes losses due to ventilation, transmission, and thermal bridges. The analysis shows how the envelope as a whole does not comply with the current values imposed by the local regulations. The thermal transmittance values, found on the basis of the hypotheses in the current situation, were approximately 1 W/m2K compared to the 0.26 W/m2K required by law for the roof slab and over 1 W/m2K compared to the 0.29 W/m2K required for the ground slab.

On the other hand, the vertical closures corresponding to the pillars were assumed to be the most critical points of the envelope. As a matter of fact, their transmittance values were ten times higher than those required, equal to 2.94 W/m2K compared to 0.29 W/m2K required by law. The cold cut aluminum type windows with single glass are responsible for almost 60% of heat losses, with a Uw transmittance value of approximately 6 W/m2K compared to the 1.80 W/m2K required by law, calculated for the largest window since it is the most critical and at the same time most widespread typology.

### 3.4. Lighting

In Ex Bastogi, each 25 square meter module, the basic component of each apartment, has a glazed area of approximately 7 square meters, with a single SW or NE exposure in the buildings located along the SE–NW axis, and with a single SE or NW exposure in buildings located along the SW–NE axis. The indoor natural and artificial lighting conditions of the single apartments in relation to the exposure provided by the Relux software, highlighted how the modules with NE exposure represent the most disadvantaged condition, while the modules exposed to SW are the most privileged ones. The apartments facing NE, despite the large glazed surfaces, are poorly lit, especially in the back. The modules exposed to SW benefit from a much better natural light, which indeed reaches about 200 lx even in the areas further back (Figure 5).

### 3.5. Ventilation

The microclimatic analyses carried out with ENVI-met highlight that the most significant problems are related to the summer season. The area, in general, is hit by fairly weak winds coming mainly from SE with maximum values of about 2.20 m/s (the simulations are carried out at a height of 1 m), yet, given the high percentage of impermeable cemented surfaces, the area is subject to a strong overheating phenomenon. In winter the winds, coming mainly from N/NW with maximum speed values similar to the summer ones, are not particularly cold and violent but, nevertheless, they significantly affect indoor comfort given the poor building performance.

The indoor air flow simulations carried out with Flow Design have shown how air exchange is inadequate in all rooms due to the absence of natural cross ventilation. In particular, in the NE-facing apartments, the exposure facilitates the entry of cold winter winds, exacerbating the overall indoor comfort conditions. In summer, the lack of natural ventilation combined with the significant thermal loads, such as lighting devices that dissipate 90% of electricity in heat, electronic devices, and household appliances, significantly contributes to indoor overheating.

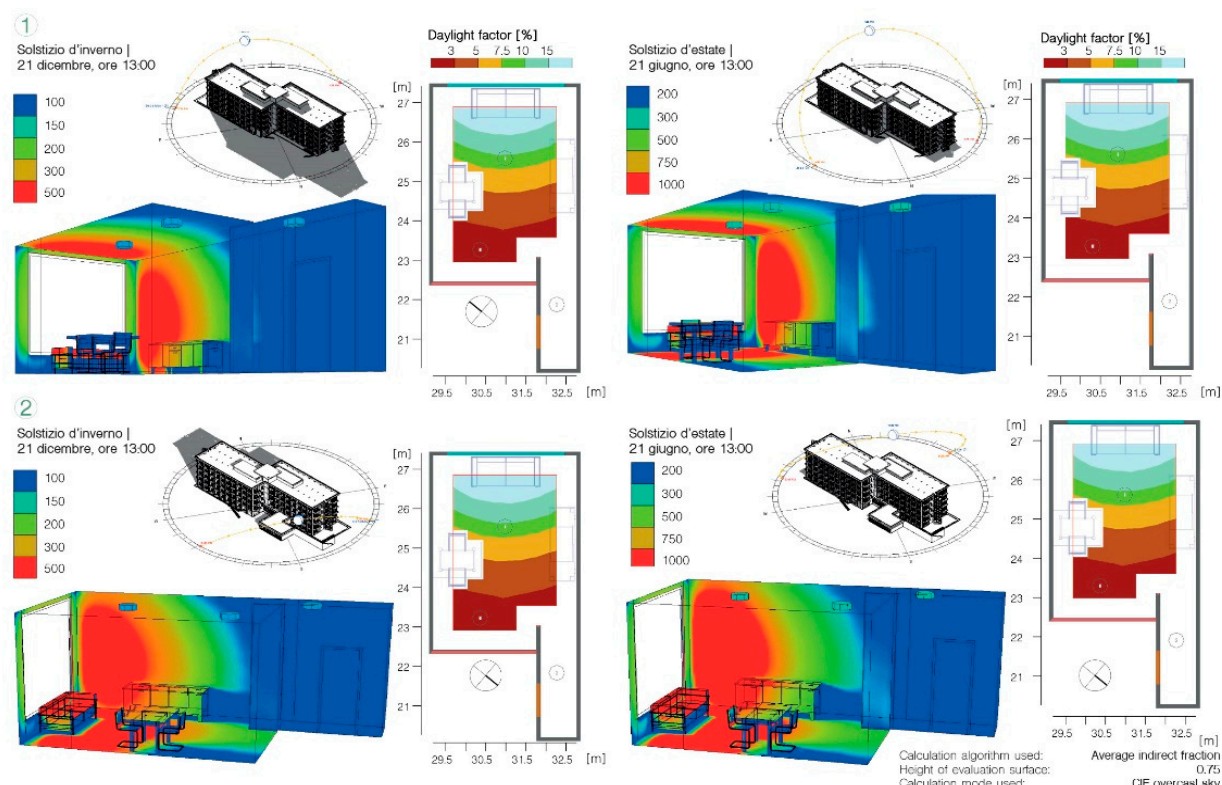

**Figure 5.** Indoor lighting analysis—Relux.

## 4. Discussion

The study revealed that non-smokers, who reported having a diagnosis of asthma, live in poor-quality homes, with humidity and/or drafts problems. The same issues were highlighted by two out of three respondents diagnosed with COPD and non-smokers and by 44.4% of subjects with joint diseases. Furthermore, considering the total number of interviewees, what stands out is that the most recurrent pathologies diagnosed among those indicated in the questionnaire are osteoarthritis or arthritis (22%), chronic bronchitis, emphysema and respiratory failure (13%), and bronchial asthma (10%) (Figure 6).

With reference to the main health risk factors, in addition to those already reported above, it is possible to include exposure to tobacco smoke. Our survey shows that 48% of the people interviewed stated that they do not smoke in any room at home. The national data collected between 2016 and 2019 show that exposure to passive smoking at home is still relevant: 16 interviewees out of 100 declare that smoking is allowed in their home and abstention from smoking in the home is about 83.7% (Figure 6) [27].

It is widely known in the scientific literature that living in reduced spaces, or in any case inadequate with respect to the number of people who live there, makes it easier for smoke to pass from one room to another [28]. In our study, 48% of respondents are smokers and, of these, 45% said they smoke in every room of the house. Furthermore, 22% of smokers reported living with at least two other people. These data confirm the possibility of exposure to passive smoking at home even for non-smokers, a condition worsened by the inadequacy of spaces.

Regarding the risk of having domestic accidents, our result is higher than the national one recorded between 2016 and 2019, which was 7% [29]. In our study, 5.7% of respondents reported having an injury at home that required medical attention (Figure 6). At the national level the percentage is only 3% [29].

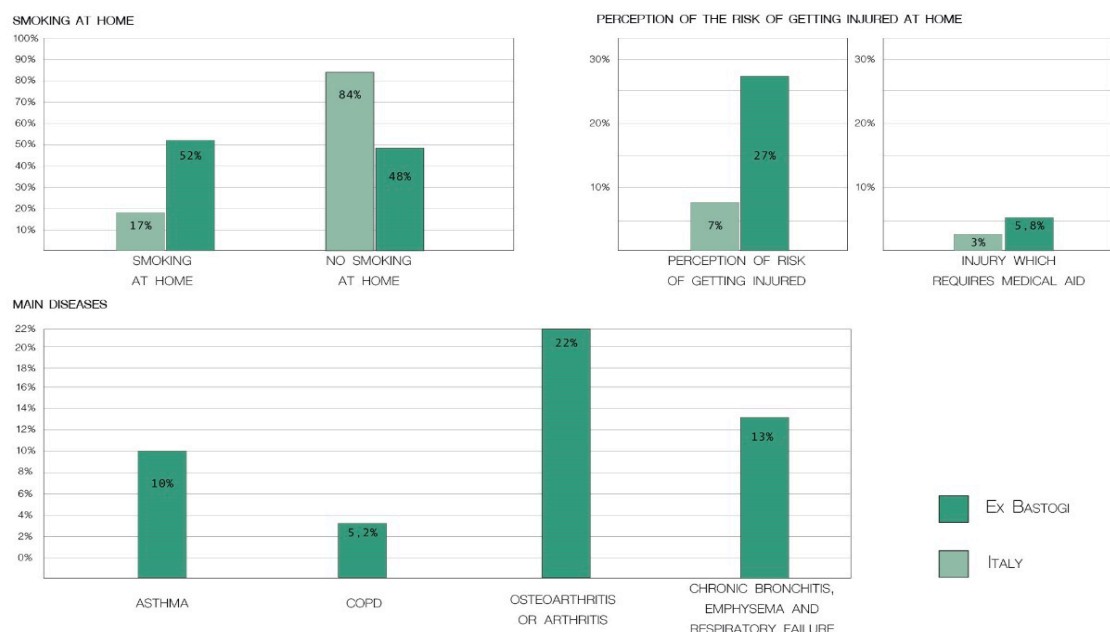

**Figure 6.** Health questionnaire outcome.

### 4.1. Spatial Comfort

The project data—number of dwellings and dimensional aspects—were compared with the effectively settled inhabitants, which are about 2000, according to the estimates of the operators and the inhabitants themselves. This comparison between the actual inhabitants and the inhabitants who can be accommodated by law shows overcrowding of 190%. Furthermore, 30% of the families count 4 or more members. Comparing the type of households with the type of housing, what stands out is that 17% of the housing is inadequate for the users. In particular, the apartments suitable to accommodate 4 people or more—that is larger than 56 m$^2$ assuming the Italian legislation as a legislative reference —are insufficient (Figure 7). Since the analysis took into consideration only the surveyed population, the lack of adequate living space is certainly more serious. Over 30% of the lodgings have bedrooms smaller than 14 m$^2$ used by at least two people. This surface area is far less than that required by the Italian national legislation. An on-site inspection revealed the presence of single bedrooms obtained from rooms without openings originally intended for toilets. These rooms do not consequently meet the health and hygiene needs or the aero-illuminating ratio required by Italian law (Ministerial Decree 5 July 1975).

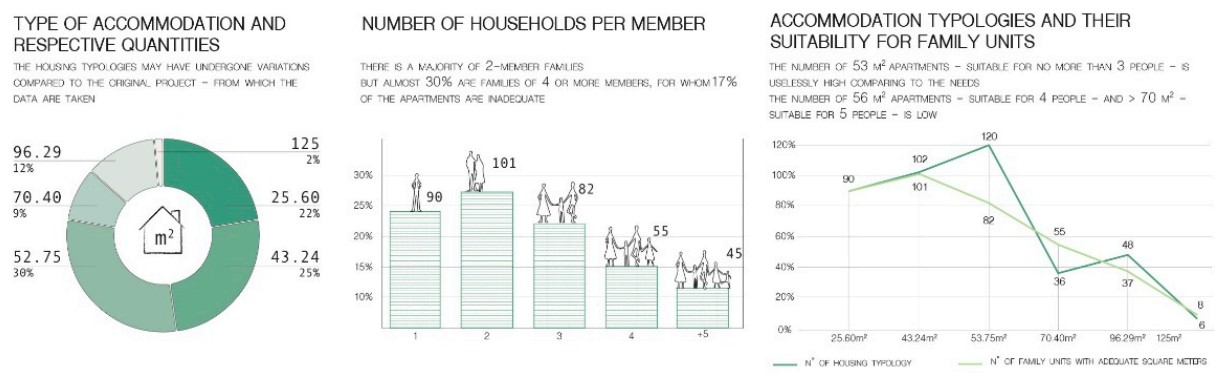

**Figure 7.** Incompatibility between inhabitants and housing typologies.

Overcrowded conditions occur when the number of occupants exceeds the capacity of available living space, measured in rooms, bedrooms, or floor space, resulting in damage to physical, mental, and social health [30,31]. The 2018 WHO report underlines how

spatial discomfort, such as overcrowding and inadequate living space, compromises the quality of living in all its aspects, from living to sleeping. The difficulty of preserving good hygiene conditions in overcrowded spaces increases the risk of domestic accidents and of disease contagion, especially respiratory and infectious ones. The same report identifies infectious diseases from close contact, such as gastroenteritis and diarrheal diseases, psychological stress, and sleep disturbances amongst the main pathologies caused by spatial discomfort. In addition, overcrowding increases exposure to risk factors associated with second-hand smoke [32].

The integration of prefabricated protruding units on the existing façade would enlarge the indoor surface of each module, thus addressing the scarce indoor and outdoor living space and simultaneously meeting the requirements of Italian law.

*4.2. Thermo-Hygrometric Comfort*

The thermal performance calculation conducted on the case study building shows that the building envelope is not able to protect the indoor environment from outdoor temperatures. According to the sunlight analysis (Figure 8), the roofs are the surfaces most affected by solar radiation at the latitude of the area under consideration. These represent a critical element during both seasons, contributing in summer to indoor overheating and in winter to thermal dispersion. On the other hand, they are configured as the most suitable surfaces for the implementation of photovoltaic systems for electricity production since they are not affected by any shadows cast by the surrounding elements. The external vertical closures exposed to SE/SW (especially those to SW) contribute in summer to the overheating of indoor spaces. Those exposed to NE/NW represent the most critical façades in terms of thermal dispersion during the winter season, as they are not affected by sunlight. Therefore, all the envelope elements will require particular attention during the regeneration design phase in terms of insulation.

Almost 50% of the inhabitants say that their apartment is not adequately heated during winter, a condition further worsened by the absence of a gas network in the entire complex. The envelope and the current plant equipment place the buildings in energy class G.

Cold air, inflaming the lungs and inhibiting circulation, makes individuals prone to respiratory diseases, such as asthma attacks, or to the contraction or worsening of diseases such as chronic obstructive pulmonary disease (BPCO) and of cardiovascular nature, including ischemic heart disease (IHD), heart diseases, and strokes [33]. In this regard, the WHO itself has indicated a minimum indoor temperature of 18 °C, which should be slightly higher for vulnerable groups including the elderly, children, and people with chronic diseases, especially cardiorespiratory ones [3,34]. Similarly, effects on health are also detectable in the case of excessively high temperatures [35,36]. A study held at Harvard University has shown how indoor temperatures, excessively hot or cold, significantly affect sleep quality with effects on mental health, such as loss of productivity and reduced cognitive function [37].

The replacement of the existing façade with a new bioclimatic skin could optimize the performance of the envelope and guarantee thermo-hygrometric comfort thanks to a better U value. In addition, the introduction of an energy storage system (solar greenhouse) at the SW of the complex, which exploits the winter solar radiation through the greenhouse effect for thermal storage, would significantly reduce heat loss in winter. The same system, in summer, could work as a buffer space further protecting the innermost rooms from hot air inlet.

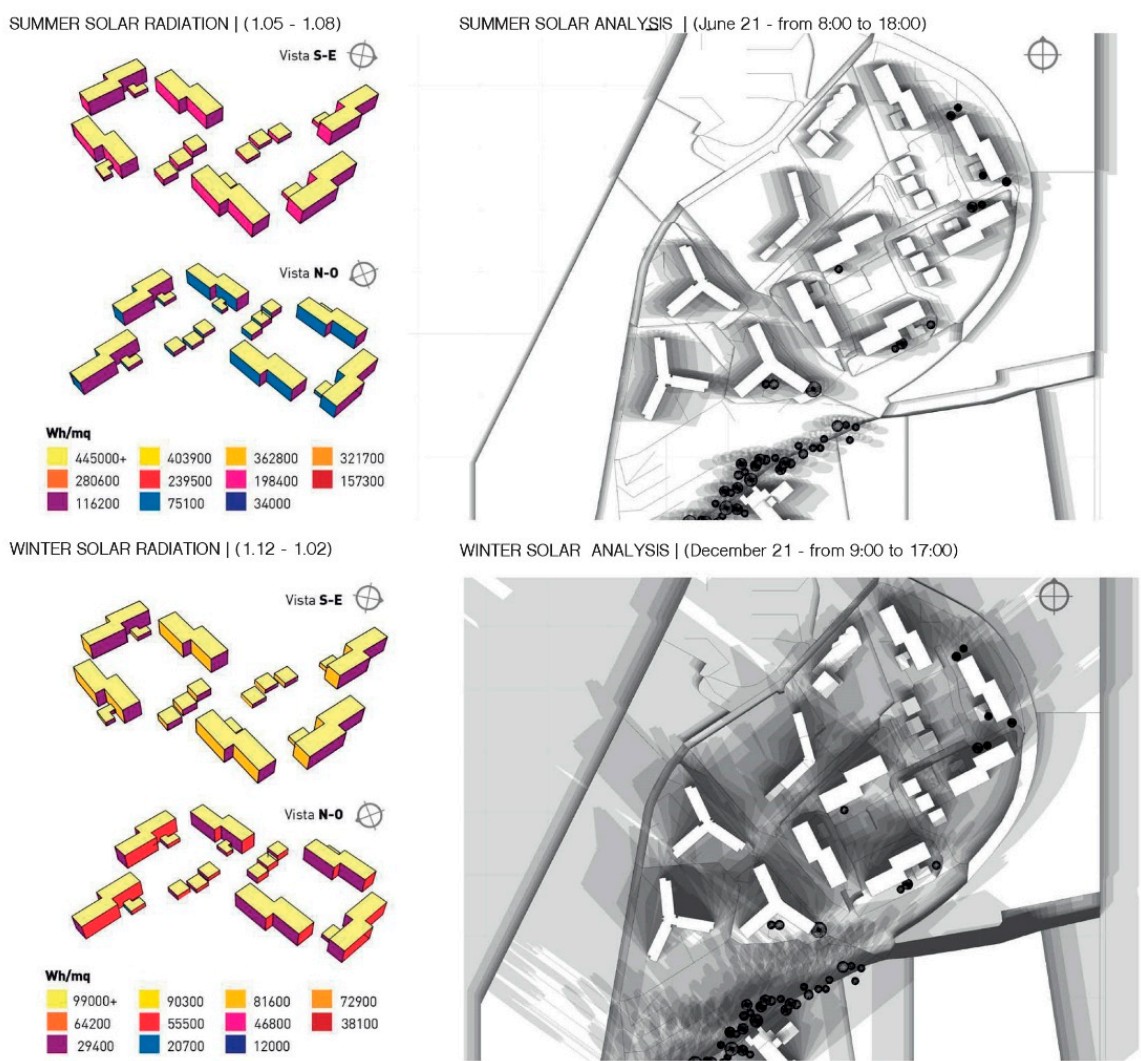

**Figure 8.** Sunlight analysis—Autodesk Ecotect Analysis.

### 4.3. Lighting

Indoor visual comfort is strongly influenced by the quantity and quality of artificial or natural lighting within a given environment at a given time. Current lifestyles, characterized by an increase in the time spent indoors, do not foster an appropriate exposure to light, which is further compromised by glass components that block a part of the spectrum of incoming light. In this regard, it has been shown that low exposure to ultraviolet (UV) rays can lead to a lack of vitamin D3 and that artificial light, even at an optimal quantity and quality, cannot replace natural light [24]. Decades of research show that artificial light is one of the factors most closely associated with sleep deprivation and disturbances, which represent risk factors for pathological conditions such as obesity, diabetes, cardiovascular diseases, hypertension, depression, heart attack, and stroke [38].

In order to maximize the amount of sunlight in the apartments composed of modules facing NE, the introduction of a protruding façade module, transparent on two sides, would allow natural light to enter not only from the NE front but also partly from SE, receiving the morning light.

### 4.4. Ventilation

Given the overall microclimatic considerations (Figure 9), it is clear how the context morphology and the specific building design, don't foster good natural ventilation of the

living spaces. Moreover it is necessary to take into account the severe state of deterioration to which the building components are prone. According to the PASSI results, almost 20% of the inhabitants complain about the presence of drafts due to inadequate and obsolete windows. Building E appears to be the most affected one since 40% of occupants report the presence of mold inside their homes.

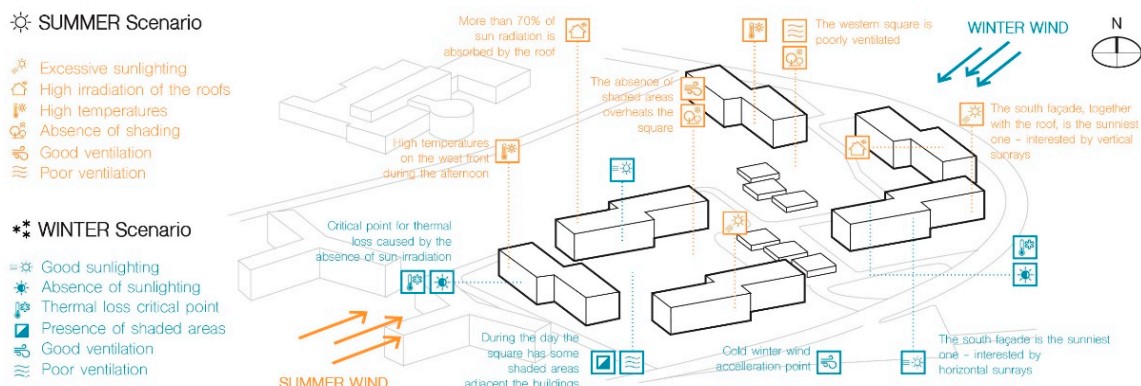

**Figure 9.** Microclimatic considerations.

The concentration of atmospheric pollutants is often two to five times higher indoors than outdoors. Good ventilation of indoor spaces can decrease the concentration of any contaminant generated by the occupant himself, such as dioxide particles of nitrogen (NO2) produced when cooking. Good ventilation contributes to the reduction of so-called sick building symptoms (SBS), such as headache, nose, throat and eye irritation, dry cough, concentration difficulties, nausea, and fatigue [37].

An enhancement of natural ventilation could be achieved through the integration of a geothermal heating/cooling system operated by wind towers. In this case, the tower collection heads should be placed where airspeed accelerations occur both in summer and in winter. The effectiveness of the system is confirmed by the simulations performed with Flow Design assuming a simplified model of the basic module. The model is equipped with an inlet device, simulating air coming from the underground ventilation ducts, and a stale air extraction device, associated with an extraction shaft with ascending flow to draw up stale air flows from the apartments.

## 5. Conclusions

Let us briefly outline the main critical issues of the research before drawing conclusions. Our limited knowledge of reality, which partially depends on our perceptive cultural system, the Zeitgeist [39], comes to light when elaborating choices, analyses, and methodology in dealing with the main problems identified by the inhabitants. The same issue also affects the questionnaire answers, focus groups, and community consultations. Another aspect that should not be underestimated is the possible failure of participatory processes. In order to obtain positive effects in a given area, triggering a participatory planning process is not necessarily enough. Since spatial and social segregation nourish each other, generating a phenomenon of ghettoization, Ex Bastogi appears to be a sort of enclave within the city, strongly marked by social vulnerability and structural violence, by housing precariousness and unauthorized property due to the legal indefiniteness of ownership and management, by the absence of socialization spaces of a solidarity network and of public utilities at the ground floor, which are all aspects that strongly increase the perception of general insecurity. Not only is there a scant willingness for dialog and low trust on the part of the inhabitants, but the incompatibility of the juridical-administrative structure with the logic of participation, which implies resistance to change, can also endanger the effectiveness and the very realization of the participatory process [40]. The lack of power entrusted to those who should directly benefit from the process, the disregard of the

created expectations, and the mismanagement of conflicts or the participant dissatisfaction can contribute to triggering a sense of helplessness in the community and a deterioration of the social capital [41]. A direct consequence of the failure of the participatory process is a sense of frustration among the participants, so that a new engagement proposal is more likely to be frowned upon and eventually rejected. Conversely, a tangible institutional commitment, aimed at structurally improving the quality of life of the inhabitants, together with interventions to strengthen social cohesion, contribute to the creation of a positive environment, of community empowerment, and of greater trust in public institutions.

The joint analysis of the questionnaires and comfort parameters shows that acting on dwellings is essential to contain or avoid negative health outcomes. Decent homes trigger a virtuous circle that contributes both to the improvement of the health conditions of the residents and to the reduction of health and energy costs. Furthermore, improving housing conditions and reducing health risks is extremely important since it contributes to the achievement of SDGs 3 and 11 [42].

According to the UN, the environment around us can affect our habits and our lifestyles. For this reason, the improvement of living spaces is one of the essential goals to be achieved by 2030 and urban planning obviously plays a crucial role. In order to protect the population's health and the sustainability of the social welfare systems, it is crucial to adopt an urban health approach, based on building concerted alliances between the health sector and other sectors such as architecture, engineering, and urban planning and research [43].

The case study offered a picture of the recurring problems associated with certain technical, dimensional, and distributional aspects. For each case study, a design simulation was carried out to assess, on one hand, the current state and, on the other, the possible improvements regarding not only the adaptation to the minimum urban standard of habitability, energy efficiency, and internal comfort, but also the economic feasibility of the improvements. One of the main efficiency measures is the replacement of the existing façade with a new modular three-dimensional bioclimatic skin, which could optimize the performance of the envelope and guarantee thermo-hygrometric comfort thanks to a new U < 0.29 W/m2K (opaque wall) and U < 0.8 W/m2K (transparent closures). The juxtaposition of a series of modular units, corresponding to the existing internal module, backed against the façade, would simultaneously solve problems related to poor lighting and under-sizing. The units, appropriately diversified on a technical level according to exposure, offer new indoor or outdoor living spaces that can vary inside in relation to the function of the room to which they are added (living-room, bedroom, etc.) and to the specific needs of the occupants. The use of these prefabricated modules would allow intervention from the outside only, without interfering with the occupants' daily life. As for the opaque NW and SE envelope, the introduction of external insulation would ensure thermo-hygrometric comfort thanks to a U < 0.29 W/m2K value.

The computer simulations compared the existing scenario and the improved one in order to verify the benefits achievable through the requalification process. The following are the main benefits of this operation:

- Technological-morphological innovation realized with a low-cost intervention on the existing buildings through the implementation of passive solar systems, greenhouses in the south, and thermal buffer spaces in the north;
- Thermal insulation of the architectural envelope and thermal cut window frames equipped with double glazed windows;
- Energy savings thanks to the optimization of thermal insulation and passive solar gains;
- Replacement of the existing electrical system with a high-efficiency LED lighting system in the common areas;
- Installation of an air-water heat pump (powered by electricity) for heating and cooling;
- Improvement of indoor life quality through an increase in the apartment's surface and an internal redistribution of undersized apartment units;

- Improvement of air quality and psycho-perceptual well-being thanks to the increase of natural ventilation fostered by wind towers;
- Integration of a photovoltaic system for electric renewable energy production to be used exclusively for cooking needs by replacing the gas cylinders with induction plates; a minimum of 27 panels with nominal power of 350 W each is required, taking into consideration the maximum annual production in Rome (1 kW plant) equal to 1250 kWh/year and assuming the location of the panels on the rooftop;
- Energy savings through the introduction of a more efficient thermal energy production system;
- Increase in safety levels, with consequent energy savings, by replacing the current energy-consuming, obsolete, and dangerous plant equipment, including the domestic equipment (such as gas cylinders in kitchens or damaged and deteriorated and leaking pipes);
- Management and recovery of water resources through the reconfiguration of the roof stratigraphy and the implementation of a new water collection system (descendants, storage, distribution network, etc.); the collected, filtered, and purified rainwater can be reused for house cleaning, toilet drains, laundry, and irrigation, achieving up to 50% savings on potable water consumption.

Conversely, a tangible institutional commitment aimed at structurally improving the quality of life of the inhabitants, together with interventions to strengthen social cohesion, can create a favorable climate, processes of community empowerment, and greater trust in public institutions.

Due to COVID 19, it was not possible to carry out air quality measurements (AIQ) inside the apartments; however, future research studies intend to monitor and improve indoor air quality in an innovative way with equipment connected to a network of wireless detection sensors located at fixed points that will provide information in real time about the environment and potentially volatile pollutants.

**Author Contributions:** Conceptualization by A.B., S.I. and M.M.; L.C. and A.C. analyzed the data collected; L.C. and A.C. designed and performed the microclimate and lighting simulations; L.C. and A.C. wrote the original draft; A.A. and M.E. took care of the PASSI related methodology, data collection, and analysis; images and figures were made by L.C. and A.C.; writing—review and editing was conducted by A.B. and S.I. All authors have read and agreed to the published version of the manuscript.

**Funding:** This research was funded by MIUR (Ministry of University and Research) 20 delle attività di base di ricerca della Prof. Alessandra Battisti, grant number RM120172B3B15DB9" and "The APC was funded by Sapienza University of Rome".

**Institutional Review Board Statement:** Not applicable.

**Informed Consent Statement:** Informed consent was obtained from all subjects involved in the study.

**Conflicts of Interest:** The authors declare no conflict of interest.

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
