# Peer review of "Urban Health: Assessment of Indoor Environment Spillovers on Health in a Distressed Urban Area of Rome"

_sustainability, doi:10.3390/su13105760_

Round 1

Reviewer 1 Report

Line 15-16 – In the abstract, the sentence “It is notable that indoor environment quality plays a crucial role in guaranteeing health, especially if we consider that  people spend  more  than  90%  of  their  time  indoor,  percentage  that increases  for  people  on  low  income..”, the word *a* should be placed before the word percentage to improve sentence flow.

The authors address a pertinent topic of indoor environment spillovers on health and demonstrate the significance of their topic by describing how indoor environments can be a greater threat for health than outdoor environments.

Authors aim to address a gap in the literature by asserting that the perception of indoor comfort should be broadened to consider a “multisensorial experience that includes five subsystems: thermal comfort, visual comfort, indoor air quality, acoustic comfort and spatial comfort”.

The rationale for choosing to assess the regeneration of the Ex Bastogi complex should be further emphasized.  Authors should further describe the need to study this particular complex.  That is, what is it about this particular complex that warrants the need for further study?  Further, are the conditions of emergency housing in Rome a situation that requires additional attention?

Lines 33-34 – The sentence “according to global estimates, indoor pollution represents the third cause of malaise and illness” should say third *leading* cause.

Lines 72-23 – In “Prevention Departments of the ASL”, is ASL an acronym?  If so, it should be written when first introduced? 

Lines 117-119 – In “the complex, consisting of six buildings erected in the 80s as a residence for Alitalia employees, was subsequently assigned by the municipality of Rome to deal with the housing emergency”, the authors should elaborate on what is the housing emergency that they are referring to.

For sampling and data collection, were participants provided any incentives for participation?  What were the questionnaire items (questions asked to participants)? Was any socioeconomic status data available for participants (e.g., income, education, occupation)?

Overall, this is an insightful, pertinent, and comprehensive study.  Attending to the clarifying questions posed may help to improve the quality of the paper.

Author Response

Authors reply: We would like to thank you for the input.  We have further described why this complex was chosen as object of study rather than other situations in Rome and we have described its singularity within the roman context: “Overcrowding of the buildings, inadequate accommodation and the perception of a transience condition extended over the years without any hope of being transferred by the municipality to the ERP, are all factors that contribute to worsen the condition of social hardship of the inhabitants. Despite a pronounced social vulnerability can be observed in several urban areas around the world, the Ex Bastogi complex stands out due to the fact that the buildings’ legal definition and status to this day, about 30 years after its foundation is devoid of a definitive framework.”

Line 15-16 – We have placed the word *a* should before the word percentage to improve sentence flow in the sentence “It is notable that indoor environment quality plays a crucial role in guaranteeing health, especially if we consider that people spend more than 90% of their time indoor, percentage that increases for people on low income…”

Line 33-34 – We have added the word *leading* before the word cause.

Line 72-73 – Since ASL is an acronym we replaced it with its broader description: “Prevention Departments of the Local Health Authority”.

Line 117 -119 (now 126 -129) – We have specified the housing emergency situation to which we refer: “with the aim of assigning a home to families in need. These housing assignments were offered on a temporary basis (a few months at most), pending final assignment in permanent housing, under the Italian acronym ERP (Public Residential Housing)”.

For sampling and data collection, participants weren’t provided any incentives for participation. The questionnaire items have been further explained.

SECTION 1: Health condition and perceived quality of life

1.5 Has a doctor ever diagnosed or confirmed one or more of the following diseases to you? (diabetes; kidney failure; bronchial asthma; chronic bronchitis, emphysema, respiratory failure; stroke or cerebral ischemia; myocardial infarction, cardiac ischemia or coronary heart disease; other heart diseases; tumors; chronic liver disease, cirrhosis; osteoarthritis or arthritis)

SECTION 3: Smoking habits

3.3 Do you currently smoke cigarettes? (yes; no)

3.10 Which of the following situations most closely resembles your smoking habits in your home? (no smoking in any room of the house; you can smoke in some rooms or at certain times or situations; you can smoke everywhere; I don't know / I don't remember)

SECTION 13: Home safety

13.1 In your opinion, what is the possibility for your household to have an injury in the home environment? (absent; low; high; very high)

13.1(b) In the past 12 months, have you had a domestic accident that required the care of your family doctor, emergency room or hospital? (yes; no; I don't know / I don't remember)

For what concerns the social, housing and cognitive questionnaire on the residents’ access to social and health services.

2.3 Who lives in the house? (only family; also outside the family)

2.4 How many square meters are there in the house?

2.5 Is your home adequately heated in winter? (yes; no)

2.6 What problems do you find in your home? (running water, electricity, gas, drafts, humidity / molds)

2.7 To reach the house, or in the house itself, are there any of these obstacles that can limit or prevent the movement of people with mobility difficulties? (stairs to the house not provided with slides or lifts; steps inside the house; small interior spaces; doors limited in width; bathroom not accessible; other)

We have integrated data we had collected and analysed regarding the socioeconomic status data of participants (e.g., income, education, occupation): “As regards the level of education, 2% of respondents have no qualifications, 9% have obtained an elementary degree, 53% have obtained an average degree, 29% have a diploma or a secondary school qualification, 6% have a degree. At the moment of the interview, 40.2% of the respondents stated that they were continuously employed and 73.1% stated that they had a lot or some difficulties in coping with their normal monthly expenses.”

Thank you for the encouraging words.

Reviewer 2 Report

none

Author Response

We would like to thank you for the input.  We have integrated some references

Reviewer 3 Report

  1. In the Introduction, please review the literature on the topic "cities & health". You mentioned a few studies but it is not enough.
  2. In the Introduction, please introduce your hypotheses and/or theories.
  3. In the Results, please use the theories you mentioned before.
  4. In the Discussions, please compare your research with similar researches and describe the future research.
  5. Please remove or reason the selfcitations.

Author Response

Thanking the reviewer, as suggested we have made the following revisions:

  • We have further extended the literature about “cities and health” (ex 117-122)
  • We have introduced our hypotheses and/or theories in the introduction: “The intent of the study is to assess the impact that social gradients can have on the lifestyle and well-being of a population, especially in such a marginal area, where housing conditions strongly contribute to an increase in mortality and morbidity amongst the most vulnerable and discriminated population groups.” (line 154 -162)
  • We have described future research in the final part of the paper. “Due to COVID 19 it wasn’t possible to carry out air quality measurements (AIQ) inside the apartments, however future research studies intend to monitor and improve indoor air quality in an innovative way with equipment connected to a network of wireless de-tection sensors located in fixed points that will provide information in real time about the environment and potentially volatile pollutants.” (line 686-690)

The self-citations are due to the fact that this study is part of a broader research project and it is the natural continuation and development of a previous study to which we must refer in order to make the overall outcome clear to the reader.  Furthermore, the only data available concerning the area under examination are offered by the broader research project which we refer throughout the paper.

Round 2

Reviewer 1 Report

The authors have done well to respond to the requested feedback.  The revised manuscript is clearer and appears suitable for publication.